# Robust Goal Recognition with Operator-Counting Heuristics

**Felipe Meneguzzi[1], André Grahl Pereira[2], and Ramon Fraga Pereira[1]**
[1]Pontifical Catholic University of Rio Grande do Sul (PUCRS), Brazil
[2]Federal University of Rio Grande do Sul (UFRGS), Brazil
felipe.meneguzzi@pucrs.br, ramon.pereira@edu.pucrs.br
agpereira@inf.ufrgs.br

## Abstract

Goal recognition is the problem of inferring the correct goal towards which an agent executes a plan, given a set of goal hypotheses, a domain model, and a (possibly noisy) sample of the plan being executed. This is a key problem in both cooperative and competitive agent interactions and recent approaches have produced fast and accurate goal recognition algorithms. In this paper, we leverage advances in operator-counting heuristics computed using linear programs over constraints derived from classical planning problems to solve goal recognition problems. Our approach uses additional operator-counting constraints derived from the observations to efficiently infer the correct goal, and serves as basis for a number of further methods with additional constraints.

## Introduction

Agents that act autonomously on behalf of a human user must choose goals independently of user input and generate plans to achieve such goals (Meneguzzi 2009). When such agents have complex sets goals and require interaction with multiple agents that are not under the user's control, the resulting plans are likely to be equally complex and non-obvious for human users to interpret (Chakraborti *et al.* 2018). In such environments, the ability to accurately and quickly identify the goals and plans of all involved agents is key to provide meaningful explanation for the observed behavior. Goal recognition is the problem of inferring one or more goals from a set of hypotheses that best account for a sequence of observations, given a fixed initial state, a goal state, and a behavior model of the agent under observation. Recent approaches to goal recognition based on classical planning domains have leveraged data-structures and heuristic information used to improve planner efficiency to develop increasingly accurate and faster goal recognition algorithms (Martín *et al.* 2015; Pereira *et al.* 2017). Specifically, Pereira *et al.* (2017) use heuristics based on planning landmarks (Hoffmann *et al.* 2004) to accurately and efficiently recognize goals in a wide range of domains with various degrees of observability and noise. This approach, however, does not deal with noise explicitly, relying on the implicit necessity of landmarks in valid plans for goal hypotheses to achieve com-

petitive accuracy with other methods (Sohrabi *et al.* 2016; Ramírez and Geffner 2010), while increasing the number of recognized goals (spread).

Thus, goal recognition under partial observability (i.e., missing observations) in the presence of noisy observation is a difficult problem to address while achieving both reasonable recognition time (i.e., a few seconds), high accuracy and low spread. In this paper, we address these limitations by leveraging recent advances on operator-counting heuristics (Pommerening *et al.* 2014; van den Briel *et al.* 2007). Operator-counting heuristics provide a unifying framework for a variety of sources of information from planning heuristics (Hoffmann *et al.* 2004) that provide both an estimate of the total cost of a goal from any given state and and indication of the actual operators likely to be in such plans. This information proves to be effective at differentiating between goal hypotheses in goal recognition.

Our contributions are threefold. First, we develop three, increasingly more accurate goal recognition approaches using operator-counting heuristics.Second, we empirically show that these heuristics are very effective at goal recognition, overcoming existing approaches in almost all domains in terms of accuracy while diminishing the spread of recognized goals. Such approaches are substantially more effective for noisy settings. Third, we discuss a broad class of operator-counting heuristics for goal recognition that can use additional constraints to provide even finer handling of noise and missing observations.

## Background

We review the key background for the approaches we develop in this paper. First, the recognition settings we assume for our approach follows the standard formalization of goal recognition as planningSecond, while there is substantial literature on linear programming heuristic unified on the operator-counting framework, we focus on the specific types of operator-counting constraints we actually use in our experimentation.

### Planning and Goal Recognition

Planning aims to find a sequence of actions that transforms an initial state into a goal state. Next, we formally define each of these elements.

**Definition 1** (**Predicates and State**). *A predicate is denoted by an n-ary predicate symbol p applied to a sequence of zero or more terms ($\tau_1$, $\tau_2$, ..., $\tau_n$) – terms are either constants or variables. We refer to grounded predicates that represent logical values according to some interpretation as facts, which are divided into two types: positive and negated facts, as well as constants for truth ($\top$) and falsehood ($\bot$). A state S is a finite set of positive facts $f$ that follows the closed world assumption so that if $f \in S$, then $f$ is true in S. We assume a simple inference relation $\models$ such that $S \models f$ iff $f \in S$, $S \not\models f$ iff $f \notin S$, and $S \models f_1 \wedge ... \wedge f_n$ iff $\{f_1, ..., f_n\} \subseteq S$.*

**Definition 2** (**Operator and Action**). *An operator $a$ is represented by a triple $\langle name(a), pre(a), eff(a)\rangle$: $name(a)$ represents the description or signature of $a$; $pre(a)$ describes the preconditions of $a$, a set of predicates that must exist in the current state for $a$ to be executed; $eff(a)$ represents the effects of $a$. These effects are divided into $eff(a)^+$ (i.e., an add-list of positive predicates) and $eff(a)^-$ (i.e., a delete-list of negated predicates). An action is a ground operator instantiated over its free variables.*

**Definition 3** (**Planning Domain**). *A planning domain definition $\Xi$ is represented by a pair $\langle \Sigma, \mathcal{A} \rangle$, which specifies the knowledge of the domain, and consists of a finite set of facts $\Sigma$ (e.g., environment properties) and a finite set of actions $\mathcal{A}$.*

**Definition 4** (**Planning Instance**). *A planning instance $\Pi$ is represented by a triple $\langle \Xi, \mathcal{I}, G \rangle$, where $\Xi = \langle \Sigma, \mathcal{A} \rangle$ is the domain definition; $\mathcal{I} \subseteq \Sigma$ is the initial state specification, which is defined by specifying values for all facts in the initial state; and $G \subseteq \Sigma$ is the goal state specification, which represents a desired state to be achieved.*

**Definition 5** (**Plan**). *An s-plan $\pi$ for a planning instance $\Pi = \langle \Xi, \mathcal{I}, G \rangle$ is a sequence of actions $\langle a_1, a_2, ..., a_n \rangle$ that modifies a state $s$ into a state $S \models G$ in which the goal state $G$ holds by the successive execution of actions in $\pi$ starting from $s$. An $\mathcal{I}$-plan is just called a plan. A plan $\pi^*$ with length $|\pi^*|$ is optimal if there exists no other plan $\pi'$ for $\Pi$ such that $|\pi'| < |\pi^*|$.*

A goal recognition problem aims to select the correct goal of an agent among a set of possible goals using as evidence a sequence of observations. These observations might be actions executed by the agent or noise observation which are part of a valid plan but are not executed by the agent.

**Definition 6** (**Observation Sequence**). *An observation sequence $O = \langle o_1, o_2, ..., o_n \rangle$ is said to be satisfied by a plan $\pi = \langle a_1, a_2, ..., a_m \rangle$, if there is a monotonic function $f$ that maps the observation indices $j = 1, ..., n$ into action indices $i = 1, ..., n$, such that $a_{f(j)} = o_j$.*

**Definition 7** (**Goal Recognition Problem**). *A goal recognition problem is a tuple $T_{GR} = \langle \Xi, \mathcal{I}, \mathcal{G}, O \rangle$, where: $\Xi = \langle \Sigma, \mathcal{A} \rangle$ is a planning domain definition; $\mathcal{I}$ is the initial state; $\mathcal{G}$ is the set of possible goals, which include a correct hidden goal $G^*$ (i.e., $G^* \in \mathcal{G}$); and $O = \langle o_1, o_2, ..., o_n \rangle$ is an observation sequence of executed actions, with each observation $o_i \in \mathcal{A}$, and the corresponding action being part of a valid*

plan $\pi$ (from Definition 5) that transitions $\mathcal{I}$ into $G^*$ through the sequential execution of actions in $\pi$.

**Definition 8** (**Solution to a Goal Recognition Problem**). *A solution to a goal recognition problem $T_{GR} = \langle \Xi, \mathcal{I}, \mathcal{G}, O \rangle$ is a nonempty subset of the set of possible goals $\mathbf{G} \subseteq \mathcal{G}$ such that $\forall G \in \mathbf{G}$ there exists a plan $\pi_G$ generated from a planning instance $\langle \Xi, \mathcal{I}, G \rangle$ and $\pi_G$ is consistent with O.*

## Operator-Counting Heuristics

Recent work on linear programming (LP) based heuristics has generated a number of informative and efficient heuristics for optimal-cost planning (van den Briel *et al.* 2007; Pommerening *et al.* 2014; Bonet 2013). These heuristics rely on constraints from different sources of information that every plan $\pi$ (Definition 5) must satisfy. All operator-counting constraints contain variables of the form $\mathsf{Y}_a$ for each operator $a$ such that setting $\mathsf{Y}_a$ to the number of occurrences of $a$ in $\pi$ satisfies the constraints. In this paper we adopt the formalism and definitions of Pommerening *et al.* for LP-based heuristics[1].

**Definition 9** (**Operator-Counting Constraints**). *Let $\Pi$ be a planning instance with operator set $\mathcal{A}$ and let $s$ be one of tis states. Let $\mathcal{Y}$ be a set of non-negative real-valued and integer variables, including an integer variable $\mathsf{Y}_a$ for each operator $a \in \mathcal{A}$ along with any number of additional variables. The variables $\mathsf{Y}_a$ are called operator-counting variables. If $\pi$ is an s-plan, we denote the number of occurrences of operator $a \in \mathcal{A}$ in $\pi$ with $\mathsf{Y}_a^\pi$. A set of linear inequalities over $\mathcal{Y}$ is called an operator counting constraint for $s$ if for every s-plan there exists a feasible solution with $\mathsf{Y}_a = \mathsf{Y}_a^\pi$ for all $a \in \mathcal{A}$. A constraint set for $s$ is a set of operator-counting constraints for $s$ where the only common variables between constraints are the operator-counting constraints.*

**Definition 10** (**Operator-Counting Integer-Linear Program**). *The operator-counting integer program $IP_C$ for constraint set $C$ aims to minimise*

$$\sum_{a \in \mathcal{A}} cost(a)\mathsf{Y}_a \text{ subject to } C$$

*where operator-counting linear program $LP_C$ is the LP-relaxation of $IP_C$.*

**Definition 11** (**IP and LP Heuristic**). *Let $\Pi$ be a planning instance, and let $\mathcal{C}$ be a function that maps states $s$ of $\Pi$ to constraint sets for $s$. The $IP$ heuristic $h_\mathcal{C}^{IP}(s)$ is the objective value of the integer program $IP_{\mathcal{C}(s)}$. The LP heuristic $h_\mathcal{C}^{LP}(s)$ is the objective value of the linear program $LP_{\mathcal{C}(s)}$. Infeasible IPs/LPs are treated as having $\infty$ objective value.*

While the framework from Pommerening *et al.* 2013 unifies many types of constraints for operator-counting heuristics, we rely on three types of constraints for our goal recognition approaches: landmarks, state-equations, and post-hoc optimization. Planning landmarks consist of actions (alternatively state-formulas) that must be executed (alternatively

---

[1]The only difference between their formalism and ours is that we refer to operators/actions with an $a/\mathcal{A}$ variable name to differentiate it from the observations $o/\mathcal{O}$

made true) in any valid plan for a particular goal (Hoffmann *et al.* 2004). Thus, landmarks are necessary conditions for all valid plans towards a goal, and, as such, provide the basis for a number of admissible heuristics (Karpas and Domshlak 2009) and as conditions to strengthen existing heuristics (Bonet 2013). Importantly, planning landmarks form the basis for the current state-of-the-art goal recognition algorithms (Pereira *et al.* 2017; Pereira and Meneguzzi 2018). *Disjunctive action landmarks* (Zhu and Givan 2003) for a state $s$ are sets of actions such that at least one action in the set must be true for any $s$-plan, and make for a natural operator-counting constraint.

**Definition 12** (**Landmark Constraints**). *Let $L \subseteq \mathcal{A}$ be a disjunctive action landmark for state $s$ of task $\Pi$. The landmark constraint $c_{s,L}^{\mathrm{lm}}$ for $L$ is:*

$$\sum_{a \in L} \mathsf{Y}_a \geq 1$$

Net change constraints generalize Bonet's (2013) state equation heuristic, which itself relate the planning instance in question with Petri nets that represent the transitions of state variables induced by the actions, and such that tokens in this task represent net changes to the states of the problem.

Finally, Post-hoc optimization constraints (Pommerening *et al.* 2013) use the fact that certain heuristics can rule out the necessity of certain operators from plans (and thus from the heuristic estimate). For example, Pattern Database (PDBs) heuristics (Culberson and Schaeffer 1998) create projections of the planning task into a subset of state variables (with this subset being the *pattern*), such that the heuristic can partition operators into two sets of each pattern, one that changes variables in the pattern (i.e., contributes towards transitions) and the other than does not (i.e., is non-contributing).

**Definition 13** (**Post-hoc Optimization Constraint**). *Let $\Pi$ be a planning task with operator set $\mathcal{A}$, let $h$ be an admissible heuristic for $\Pi$, and let $N \subseteq \mathcal{A}$ be a set of operators that are noncontributing in that $h$ is still admissible in a modified planning task where the cost of all operators in $N$ is set to 0.*

*Then the post-hoc optimization constraint $c_{s,h,N}^{PH}$ for $h$, $N$, and state $s$ of $\Pi$ consists of the inequality.*

$$\sum_{a \in \mathcal{A} \setminus N} cost(a) \mathsf{Y}_a \geq h(s)$$

## Goal Recognition using Operator Counts

We now bring together the operator-counting constraints into three operator-counting heuristics suitable for goal recognition, ranging from the simplest way to employ operator counts to compute the overlap between counts and observed actions, to modifying the constraints used by the operator counts to enforce solutions that agree with such observations,and finally accounting for possible noise by comparing heuristic values.

### Computing Observation Overlap Count

We start with a basic operator-counting heuristic $h(s)$, which we define to be the LP-heuristic of Def. 11 where $C$ com-

---

**Algorithm 1** Goal Recognition using the Operator Counts.

**Input:** $\Xi$ *planning domain definition*, $\mathcal{I}$ *initial state*, $\mathcal{G}$ *set of candidate goals*, and $O$ *observations*.
**Output:** *Recognized goal(s).*

1: **function** OPCOUNTRECOGNIZE($\Xi, \mathcal{I}, \mathcal{G}, O$)
2:     $Hits \leftarrow$ Initialize empty dictionary
3:     **for all** $G \in \mathcal{G}$ **do**        ▷ Compute overlap for $G$
4:         $Hits_G \leftarrow 0$
5:         $\mathcal{Y} \leftarrow$ GENERATECONSTRAINTS($\Xi, \mathcal{I}, G$)
6:         $\mathsf{Y} \leftarrow$ COMPUTEOPERATORCOUNTS($\mathcal{Y}$)
7:         **for all** $o \in O$ **do**
8:             **if** $\mathsf{Y}_o > 0$ **then**
9:                 $Hits_G \leftarrow Hits_G + 1$
10:                $\mathsf{Y}_o \leftarrow \mathsf{Y}_o - 1$
11:     **return** all $G$ s.t $G \in \mathcal{G} \wedge Hits_G = \max_G Hits_G$

---

prises the constraints generated by Landmarks (Def. 12), post-hoc optimization (Def. 13), and net change constraints as described by Pommerening *et al.* (2014). This heuristic, computed following Def. 11, yields two important bits of information for our first technique, first, it generates the actual operator counts $\mathsf{Y}_a$ for all $a \in \mathcal{A}$ from Def. 10, whose minimization comprises the objective function $h(s)$.

The heuristic values $h(s)$ of each goal candidate $G \in \mathcal{G}$ tells us about the optimal distance between the initial state $\mathcal{I}$ and $G$, while the operator counts indicate *possible* operators in a valid plan from $\mathcal{I}$ to $G$. We can use these counts to account for the observations $\mathcal{O}$ by computing the *overlap* between operators with counts greater than one and operators observed for recognition. Algorithm 1 shows how we can use the operator counts directly in a goal recognition technique. In order to rank the goal hypotheses we keep a dictionary of $Hits$ (Line 2) to store the overlap, or count the times operators counts hit observed actions. The algorithm then iterates over all goal hypotheses (Lines 3-10) computing the operator counts for each hypothesis $G$ and comparing these counts with the actual observations (Lines 7–10). We recognize goals by choosing the hypotheses whose operator counts hit the most observations (Line 11).

### Enforcing Observation Constraints

The technique of Algorithm 1 is conceptually similar to the *Goal Completion* heuristic of Pereira *et al.* (2017) in that it tries to compare heuristically computed information with the observations. However, this initial approach has a number of shortcomings in relation to their technique. First, while the landmarks themselves are enforced by the $LP$ used to compute the operator counts (and thus observations that correspond to landmarks count as hits), the overlap computation loses the ordering of the landmarks that the Goal Completion heuristic uses to account for missing observations. Second, a solution to a set of operator-constraints, i.e., a set of operators with non-negative counts may not be a feasible plan for a planning instance. Thus, these counts may not correspond to the plan that generated the observations.

While operator-counting heuristics on their own are fast and informative enough to help guide search when dealing with millions of nodes, goal recognition problems often re-

---

**Algorithm 2** Goal Recognition using Observation-Constrained Operator Counts.

---
**Input:** $\Xi$ *planning domain definition*, $\mathcal{I}$ *initial state*, $\mathcal{G}$ *set of candidate goals*, and $O$ *observations*.
**Output:** *Recognized goal(s).*
 1: **function** OPCOUNTOBSRECOGNIZE$(\Xi, \mathcal{I}, \mathcal{G}, O)$
 2:     **for all** $G \in \mathcal{G}$ **do**         ▷ Compute $h_c(\mathcal{I})$ for $G$
 3:         $\mathcal{Y} \leftarrow$ GENERATECONSTRAINTS$(\Xi, \mathcal{I}, G)$
 4:         **for all** $o \in O$ **do**
 5:             $\mathcal{Y} \leftarrow \mathcal{Y} \cup (\mathsf{Y}_o > 1)$
 6:         $\mathsf{Y} \leftarrow$ COMPUTEOPERATORCOUNTS$(\mathcal{Y})$
 7:         $H_G \leftarrow \sum_{a \in \mathcal{A}} \mathsf{Y}_a$
 8:     **return** all $G$ s.t $G \in \mathcal{G} \wedge H_G = \min_G H_G$

---

quire the disambiguation of a dozen or less goal hypotheses. Such goal hypotheses are often very similar so that the operator-counting heuristic value (i.e., the objective function over the operator counts) for each goal hypothesis is very similar, especially if the goals are more or less equidistant from the initial state.

Thus, we refine the technique of Observation Overlap by introducing additional constraints into the LP used to compute operator counts. Specifically, we force the operator counting heuristic to *only consider* operator counts that include every single observation $o \in \mathcal{O}$. The resulting LP heuristic (which we call $h_C$) then minimizes the cost of the operator counts for plans that necessarily agree with all observations. We summarize this *Observation Constraint Enforcement* approach in Algorithm 2. This technique is similar to that of Algorithm 1 in that it iterates over all goals computing a heuristic value. However, instead of computing observation hits by looking at individual counts, it generates the constraints for the operator-counting heuristic (Line 3) and adds constraints to ensure that the count of the operators corresponding to each observation is greater than one (Lines 4–5). Finally, we choose the goal hypotheses that minimize the operator count heuristic distance from the initial state (Line 8).

### Enforcement Delta

Although enforcing constraints to ensure that the LP heuristic computes only plans that do contain all observations helps us overcome the limitations of computing the overlap of the operator counts, this approach has a major shortcoming: it considers all observations as valid operators generated by the observed agent. Therefore, the heuristic resulting from the minimization of the LP might overestimate the actual length of the plan for the goal hypothesis due to noise. This may happen for one of two reasons: either the noise is simply a sub-optimal operator in a valid plan, or it is an operator that is completely unrelated to the plan that generated the observations. In both cases, the resulting heuristic value may prevent the algorithm from selecting the actual goal from among the goal hypotheses. This overestimation, however, has an important property in relation to the basic operator counting heuristic, which is that $h_C$ always dominates the operator counting heuristic $h$, in Proposition 1.

---

**Algorithm 3** Goal Recognition using Heuristic Difference of Operator Counts.

---
**Input:** $\Xi$ *planning domain definition*, $\mathcal{I}$ *initial state*, $\mathcal{G}$ *set of candidate goals*, and $O$ *observations*.
**Output:** *Recognized goal(s).*
 1: **function** DELTARECOGNIZE$(\Xi, \mathcal{I}, \mathcal{G}, O)$
 2:     **for all** $G \in \mathcal{G}$ **do**         ▷ Compute $h_\delta(\mathcal{I})$ for $G$
 3:         $\mathcal{Y} \leftarrow$ GENERATECONSTRAINTS$(\Xi, \mathcal{I}, G)$
 4:         $\mathsf{Y} \leftarrow$ COMPUTEOPERATORCOUNTS$(\mathcal{Y})$
 5:         $H_G \leftarrow \sum_{a \in \mathcal{A}} \mathsf{Y}_a$
 6:         **for all** $o \in O$ **do**
 7:             $\mathcal{Y} \leftarrow \mathcal{Y} \cup \mathsf{Y}_o > 0$
 8:         $\mathsf{Y} \leftarrow$ COMPUTEOPERATORCOUNTS$(\mathcal{Y})$
 9:         $H_{C,G} \leftarrow \sum_{a \in \mathcal{A}} \mathsf{Y}_a$
10:         $H_{\delta,G} \leftarrow H_{C,G} - H_G$
11:     **return** all $G$ s.t $G \in \mathcal{G} \wedge H_{\delta,G} = \min_G H_{\delta,G}$

---

**Proposition 1** ($h_C$ **dominates** $h$). *Let $h$ be the operator-counting heuristic from Defs. 10-11, $h_C$ be the over-constrained heuristic that accounts for all observations $o \in \mathcal{O}$, and $s$ a state of $\Pi$. Then $h_C(s) \geq h(s)$.*

*Proof.* Let $C^h$ be set of constraints used in $h(s)$, and $C^{h_C}$ be set of constraint used to compute $h_C(s)$. Every feasible solution to $C_{h_C}$ is a solution to $C_h$. This is because to generate $C_{h_C}$ we only *add* constraints to $C_h$. Thus, a solution to $C^{h_C}$ has to satisfy all constraints in $C^h$. Therefore, since we are solving a minimization problem the value of the solution for $C^h$ cannot be larger than the solution to $C^{h_C}$. $\quad\square$

The intuition here is that the operator-counting heuristic $h$ estimates the total cost of any optimal plan, regardless of the observations, while $h_C$ estimates a plan following all observations, including noise, if any. If there is no noise, the sum of the counts must agree (even if the counts are different), whereas if there is noise and assuming the noise is evenly distributed, there will be differences in all counts. Thus, our last approach consists of computing the difference between $h_C$ and $h$, and infer that the goal hypothesis for which these values are closer must be the correct goal. We call the resulting heuristic $h_\delta$ and formalize this approach in Algorithm 3. Here we compute the LP twice, once with only the basic operator-counting constraints (Line 4), and once with the constraints enforcing the observations in the operator counts (Line 8), using these two values to compute $h_\delta$ (Line 10). The algorithm then returns goal hypotheses that minimize $h_\delta$ (Line 11).

## Experiments and Results

To evaluate the effectiveness of our approaches, we implemented each of the algorithms described earlier and performed the goal recognition process over the large dataset introduced by Pereira *et al.* (2017). This dataset contains thousands of problems for goal and plan recognition under varying levels of observability for a number of traditional IPC domains (Vallati *et al.* 2018), including BLOCKS-WORLD, CAMPUS, DEPOTS, DRIVERLOG, Dockworker robots (DWR), IPC-GRID, FERRY, Intrusion Detection

(INTRUSION), KITCHEN, LOGISTICS, MICONIC, ROVER, SATELLITE, SOKOBAN, and Zeno Travel (ZENO). It also contains over a thousand problems under partial observability *and* noisy observations in the CAMPUS, IPC-GRID, INTRUSION and KITCHEN domains. The baselines of our experimentation were the original deterministic approach from Ramírez and Geffner (2009) (R&G 2009) and the recent algorithms from Pereira *et al.* (2017) ($h_{uniq}$) and Martín *et al.* (2015) (FG2015)[2]. We implemented our approaches using PYTHON 2.7 for the main recognition algorithms with external calls to a customized version of the FAST-DOWNWARD (Helmert 2006) planning system to compute the operator counts. Our customized planner returns not only the operator counts and can also introduce additional constraints before running the CPLEX 128 optimization system. We ran experiments in a single core of a 24 core Intel® Xeon® CPU E5-2620 @2.00Ghz with 48GB of RAM, with a 2-minute time limit and a 2GB memory limit.

Table 1 shows the results for the partially observable, non-noisy fragment of the dataset, whereas Table 2 shows the noisy fragment of the dataset[3]. For the noisy experiments, each set of observations contained at least two spurious actions, which, while valid for the plan, were not actually executed by the agent being observed. These results show that, while not nearly as fast as the $h_{uniq}$ approach from Pereira *et al.* with a $\theta = 0$ recognition threshold, the accuracy (Acc %) of our $h_\delta$ approach is either competitive or superior in virtually all domains (except for some levels of observability in IPC-GRID, DWR and KITCHEN), and, even for the domains where the accuracy is similar, or lower, the spread ($Sin\mathcal{G}$) of the resulting goals is consistently lower, i.e., the returned goals are unique for most problems. The accuracy of our approach, thus, consistently matches or surpasses that of R&G 2009, with a computational cost that is also often smaller than FG 2015. Importantly, the cost of all of our approaches is basically the same within each domain, regardless of the level of observability and noise, since our technique relies on a single call to a planner that computes the operator counts for a single state and then stops the planner. We argue that this is attributable to our inefficient implementation rather than the technique, for the $h_\delta$ approach, the overhead of the FAST-DOWNWARD pre-processing step is paid multiple times. Unlike R&G 2009, that uses a modified planning heuristic, and FG 2015, that builds a data structure and explores it at very high computational cost. We note that the results for noisy observations show the greatest impact of $h_\delta$ with an overall higher accuracy and lower spread across all domains but KITCHEN.

Finally, results for the KITCHEN domain stand out in our experiments in that our some of our approaches consistently show underwhelming performance both in noisy and non-noisy domains. Counter-intuitively, for this particular do-

main, the more observations we have available, the worse the performance. This seems to be a problem for all other approaches under noisy conditions, though not under incomplete observations. Moreover, since the loss of accuracy with fuller observability also occurs for the non-noisy setting, we surmise this to stem from the domain itself, rather than the algorithm's ability to handle noise, and defer investigation of this issue to future work.

## Related Work

Our work follows the traditional of goal and plan recognition as planning algorithms as defined by Ramírez and Geffner (2009; 2010). The former work yields higher recognition accuracy in our settings (and hence we chose it as a baseline), whereas the latter models goal recognition as a problem of estimating the probability of a goal given the observations. Such work uses a Bayesian framework to compute the probability of goals given observations by computing the probability of generating a plan given a goal, which they accomplish by running a planner multiple times to estimate the probability of the plans that either comply or not with the observations. Recent research on goal recognition has yielded a number of approaches to deal with partial observability and noisy observations, of which we single out three key contributions. First, Martín *et al.* (2015) developed a goal recognition approach based on constructing a planning graph and propagating operator costs and the interaction among operators to provide an estimate of the probabilities of each goal hypothesis. While their approach provides probabilistic estimates for each goal, its precision in inferring the topmost goals is consistently lower than ours, often ranking multiple goals with equal probabilities (i.e., having a large spread). Second, Sohrabi *et al.* (2016) developed an approach that also provides a probabilistic interpretation and explicitly deals with noisy observations. Their approach works through a compilation of the recognition problem into a planning problem that is processed by a planner that computes a number of approximately optimal plans to compute goal probabilities under R&G's Bayesian framework. Finally, Pereira *et al.* (2017) develop heuristic goal recognition approaches using landmark information. This approach is conceptually closer to ours in that we also compute heuristics, but we aim to overcome the potential sparsity of landmarks in each domain by using operator-count information, as well as explicitly handle noise by introducing additional constraints in heuristic $h_C$ and comparing the distance to the unconstrained $h$ heuristic.

## Conclusion and Discussion

We developed a novel class goal recognition technique based on operator-counting heuristics from classical planning (Pommerening *et al.* 2014) which, themselves rely on ILP constraints to estimate which operators occur in valid optimal plans towards a goal. The resulting approaches are competitive with the state of the art in terms of high accuracy and low false positive rate (i.e., the spread of returned goals), at a moderate computational cost. We show empirically that the overall accuracy of our best approach is sub-

---

[2]We excluded the results of (Sohrabi *et al.* 2016) from our comparison as it timed out for virtually all problems in all domains, even with a 20-minute timeout.

[3]*Timeout* indicates that approach exceeded the two-minute timeout we set for the experiments, whereas the † symbol indicates a runtime failure for most problems in the domain.

Table 1:

| # | $|\mathcal{G}|$ | % Obs | $|O|$ | $h$ Time | Acc % | $S$ in $\mathcal{G}$ | $h_c$ Time | Acc % | $S$ in $\mathcal{G}$ | $h_\delta$ Time | Acc % | $S$ in $\mathcal{G}$ | R&G 2009 Time | Acc % | $S$ in $\mathcal{G}$ | FG 2015 Time | Acc % | $S$ in $\mathcal{G}$ | $h_{uniq}$ Time | Acc % | $S$ in $\mathcal{G}$ |
|---|---|---|---|---|---|---|---|---|---|---|---|---|---|---|---|---|---|---|---|---|---|
| BLOCKS (1076) | 20.0 | 10 | 1.8 | 8.896 | 23.2% | 2.17 | 8.916 | 45.1% | 2.6 | 17.812 | 95.1% | 7.74 | 1.235 | 86.8% | 7.84 | 36.562 | 65.8% | 9.11 | 0.131 | 31.6% | 1.03 |
| | | 30 | 4.9 | 8.877 | 16.3% | 1.54 | 8.897 | 67.9% | 2.02 | 17.774 | 87.8% | 2.71 | 1.698 | 87.2% | 3.56 | 36.648 | 78.1% | 10.53 | 0.144 | 51.4% | 1.06 |
| | | 50 | 7.6 | 8.874 | 14.2% | 1.37 | 8.875 | 79.3% | 1.55 | 17.749 | 91.5% | 1.74 | 2.497 | 97.9% | 2.63 | 34.290 | 81.3% | 10.68 | 0.168 | 60.1% | 1.08 |
| | | 70 | 11.1 | 8.382 | 11.0% | 1.32 | 8.4 | 93.9% | 1.21 | 16.782 | 98.4% | 1.4 | 3.704 | 97.5% | 1.83 | 37.056 | 89.8% | 8.63 | 0.184 | 79.1% | 1.13 |
| | | 100 | 14.5 | 8.208 | 10.9% | 1.12 | 8.227 | 100.0% | 1.03 | 16.435 | 100.0% | 1.21 | 6.123 | 100% | 1.46 | 40.405 | 100.0% | 1.22 | 0.238 | 100% | 1.09 |
| CAMPUS (75) | 2.0 | 10 | 1.0 | 0.631 | 53.3% | 1.0 | 0.628 | 60.0% | 1.07 | 1.259 | 100.0% | 1.27 | 0.084 | 86.8% | 1.46 | 0.717 | 53.3% | 1.0 | 0.027 | 100% | 1.13 |
| | | 30 | 2.0 | 0.628 | 53.3% | 1.0 | 0.631 | 73.3% | 1.2 | 1.259 | 100.0% | 1.07 | 0.097 | 100% | 1.33 | 0.696 | 80.0% | 1.13 | 0.042 | 100% | 1.13 |
| | | 50 | 3.0 | 0.634 | 40.0% | 1.0 | 0.63 | 93.3% | 1.27 | 1.264 | 100.0% | 1.0 | 0.104 | 100% | 1.33 | 0.676 | 66.6% | 1.26 | 0.055 | 93.3% | 1.13 |
| | | 70 | 4.4 | 0.628 | 53.3% | 1.0 | 0.629 | 100.0% | 1.07 | 1.257 | 100.0% | 1.07 | 0.115 | 100% | 1.26 | 0.668 | 86.6% | 1.6 | 0.058 | 100% | 1.0 |
| | | 100 | 5.5 | 0.624 | 60.0% | 1.0 | 0.626 | 100.0% | 1.0 | 1.25 | 100.0% | 1.13 | 0.128 | 100% | 1.13 | 0.631 | 93.3% | 1.53 | 0.061 | 100% | 1.0 |
| DEPOTS (364) | 8.5 | 10 | 3.1 | 5.76 | 15.5% | 1.29 | 5.715 | 32.1% | 1.54 | 11.475 | 53.6% | 1.83 | 1.485 | 77.3% | 3.98 | † | † | † | 0.331 | 32.1% | 1.09 |
| | | 30 | 8.6 | 5.767 | 14.3% | 1.31 | 5.693 | 69.0% | 1.64 | 11.46 | 64.3% | 1.19 | 2.307 | 77.3% | 2.39 | † | † | † | 0.356 | 47.6% | 1.07 |
| | | 50 | 14.1 | 5.476 | 14.3% | 1.21 | 5.438 | 91.7% | 1.33 | 10.914 | 85.7% | 1.1 | 3.433 | 84.5% | 1.91 | † | † | † | 0.415 | 71.4% | 1.02 |
| | | 70 | 19.7 | 5.304 | 14.3% | 1.12 | 5.252 | 100.0% | 1.08 | 10.556 | 94.0% | 1.01 | 5.149 | 91.6% | 1.67 | † | † | † | 0.481 | 84.5% | 1.01 |
| | | 100 | 24.4 | 5.238 | 14.3% | 1.21 | 5.205 | 100.0% | 1.0 | 10.443 | 100.0% | 1.0 | 7.094 | 92.8% | 1.46 | † | † | † | 0.575 | 100% | 1.03 |
| DRIVERLOG (364) | 10.5 | 10 | 2.6 | 3.342 | 32.1% | 1.43 | 3.316 | 42.9% | 1.74 | 6.658 | 73.8% | 2.43 | 1.192 | 96.4% | 4.71 | 79.487 | 42.8% | 1.91 | 0.284 | 35.7% | 1.10 |
| | | 30 | 6.9 | 3.337 | 28.6% | 1.45 | 3.351 | 75.0% | 1.45 | 6.688 | 77.4% | 1.55 | 1.444 | 92.8% | 3.34 | 60.168 | 70.2% | 3.19 | 0.284 | 35.7% | 1.10 |
| | | 50 | 11.1 | 3.338 | 28.6% | 1.13 | 3.35 | 92.9% | 1.15 | 6.688 | 91.7% | 1.17 | 1.608 | 94.1% | 2.88 | 64.427 | 79.7% | 4.59 | 0.290 | 64.2% | 1.14 |
| | | 70 | 15.6 | 3.304 | 28.6% | 1.18 | 3.308 | 97.6% | 1.12 | 6.612 | 95.2% | 1.11 | 1.925 | 89.2% | 2.46 | 75.084 | 82.1% | 4.10 | 0.298 | 90.4% | 1.14 |
| | | 100 | 21.7 | 3.301 | 28.6% | 1.04 | 3.347 | 100.0% | 1.0 | 6.648 | 100.0% | 1.0 | 2.809 | 89.2% | 2.14 | 96.091 | 96.4% | 1.11 | 0.305 | 100% | 1.17 |
| DWR (364) | 7.3 | 10 | 5.7 | 3.604 | 38.1% | 1.6 | 3.601 | 50.0% | 1.71 | 7.205 | 56.0% | 2.19 | 1.634 | 83.3% | 4.21 | 66.496 | 92.8% | 6.38 | 0.491 | 33.3% | 1.05 |
| | | 30 | 16.0 | 3.63 | 36.9% | 1.42 | 3.579 | 81.0% | 1.42 | 7.209 | 76.2% | 1.46 | 2.977 | 80.9% | 3.34 | 54.461 | 97.6% | 6.56 | 0.518 | 51.1% | 1.05 |
| | | 50 | 26.2 | 3.611 | 36.9% | 1.04 | 3.583 | 98.8% | 1.2 | 7.194 | 84.5% | 1.15 | 4.485 | 72.6% | 2.27 | 56.255 | 98.8% | 6.27 | 0.533 | 61.9% | 1.04 |
| | | 70 | 36.8 | 3.625 | 36.9% | 1.04 | 3.569 | 100.0% | 1.02 | 7.194 | 94.0% | 1.04 | 10.432 | 70.2% | 2.04 | 65.101 | 98.8% | 6.0 | 0.540 | 78.5% | 1.03 |
| | | 100 | 51.9 | 3.581 | 35.7% | 1.0 | 3.58 | 100.0% | 1.0 | 7.161 | 100.0% | 1.0 | 25.091 | 67.8% | 1.67 | 86.459 | 100.0% | 1.0 | 0.559 | 100% | 1.01 |
| IPC-GRID (673) | 9.0 | 10 | 2.9 | 3.811 | 9.8% | 1.0 | 3.828 | 18.9% | 1.01 | 7.639 | 90.8% | 1.88 | 1.084 | 96.1% | 2.45 | Timeout | - | - | 0.220 | 62.7% | 2.34 |
| | | 30 | 7.8 | 3.871 | 9.2% | 1.0 | 3.867 | 49.7% | 1.2 | 7.738 | 94.1% | 1.25 | 1.475 | 97.3% | 1.42 | Timeout | - | - | 0.234 | 83.6% | 1.66 |
| | | 50 | 12.7 | 3.821 | 9.8% | 1.0 | 3.82 | 79.1% | 1.03 | 7.641 | 96.7% | 1.07 | 1.932 | 100% | 1.15 | Timeout | - | - | 0.245 | 90.1% | 1.18 |
| | | 70 | 17.9 | 3.902 | 9.2% | 1.0 | 3.878 | 96.1% | 1.03 | 7.78 | 94.1% | 1.05 | 2.556 | 100% | 1.05 | Timeout | - | - | 0.253 | 97.3% | 1.11 |
| | | 100 | 24.8 | 3.62 | 9.8% | 1.0 | 3.637 | 98.4% | 1.0 | 7.257 | 96.7% | 1.0 | 3.868 | 100% | 1.0 | Timeout | - | - | 0.261 | 100% | 1.0 |
| FERRY (364) | 7.5 | 10 | 2.9 | 2.683 | 39.3% | 1.65 | 2.686 | 72.6% | 2.05 | 5.369 | 100.0% | 3.17 | 0.511 | 98.8% | 3.36 | 6.659 | 91.6% | 6.65 | 0.068 | 58.3% | 1.17 |
| | | 30 | 7.6 | 2.693 | 39.3% | 1.31 | 2.686 | 94.0% | 1.48 | 5.379 | 100.0% | 1.56 | 0.677 | 100% | 1.76 | 6.801 | 100.0% | 7.57 | 0.073 | 83.3% | 1.05 |
| | | 50 | 12.3 | 2.673 | 39.3% | 1.17 | 2.671 | 97.6% | 1.2 | 5.344 | 100.0% | 1.29 | 0.794 | 100% | 1.41 | 8.296 | 100.0% | 7.57 | 0.084 | 91.6% | 1.01 |
| | | 70 | 17.3 | 2.661 | 39.3% | 1.12 | 2.673 | 100.0% | 1.08 | 5.334 | 100.0% | 1.1 | 1.202 | 98.8% | 1.14 | 10.649 | 100.0% | 7.32 | 0.092 | 100% | 1.0 |
| | | 100 | 24.2 | 2.695 | 39.3% | 1.11 | 2.708 | 100.0% | 1.07 | 5.403 | 100.0% | 1.07 | 1.693 | 100% | 1.07 | 13.625 | 100.0% | 1.07 | 0.099 | 100% | 1.0 |
| INTRUSION (465) | 15.0 | 10 | 1.9 | 4.701 | 10.5% | 1.25 | 4.713 | 27.6% | 1.81 | 9.414 | 100.0% | 2.52 | 0.724 | 100% | 2.53 | 0.475 | 89.5% | 3.18 | 0.077 | 64.7% | 1.23 |
| | | 30 | 4.5 | 4.511 | 9.5% | 1.12 | 4.518 | 80.0% | 1.4 | 9.029 | 100.0% | 1.11 | 0.804 | 100% | 1.11 | 0.476 | 90.5% | 1.88 | 0.083 | 85.7% | 1.02 |
| | | 50 | 6.7 | 4.421 | 9.5% | 1.09 | 4.424 | 94.3% | 1.12 | 8.845 | 100.0% | 1.02 | 0.888 | 100% | 1.02 | 0.496 | 94.3% | 1.45 | 0.089 | 94.2% | 1.04 |
| | | 70 | 9.5 | 4.458 | 10.5% | 1.09 | 4.453 | 97.1% | 1.13 | 8.911 | 100.0% | 1.0 | 1.012 | 100% | 1.0 | 0.637 | 99.1% | 1.05 | 0.093 | 94.2% | 1.0 |
| | | 100 | 13.1 | 4.419 | 8.9% | 1.13 | 4.413 | 100.0% | 1.0 | 8.832 | 100.0% | 1.0 | 1.257 | 100% | 1.0 | 0.828 | 100.0% | 1.04 | 0.098 | 100% | 1.0 |
| KITCHEN (75) | 2.0 | 10 | 1.3 | 0.801 | 53.3% | 1.0 | 0.807 | 53.3% | 1.0 | 1.608 | 100.0% | 1.87 | 0.085 | 100% | 1.86 | 0.373 | 100.0% | 1.86 | 0.002 | 100% | 1.33 |
| | | 30 | 3.5 | 0.789 | 26.7% | 1.0 | 0.785 | 33.3% | 1.07 | 1.574 | 100.0% | 1.33 | 0.097 | 100% | 1.33 | 0.360 | 100.0% | 1.33 | 0.003 | 100% | 1.33 |
| | | 50 | 4.0 | 0.792 | 46.7% | 1.0 | 0.802 | 53.3% | 1.07 | 1.594 | 93.3% | 1.33 | 0.104 | 100% | 1.46 | 0.392 | 100.0% | 1.33 | 0.006 | 100% | 1.33 |
| | | 70 | 5.0 | 0.795 | 46.7% | 1.0 | 0.787 | 66.7% | 1.13 | 1.582 | 80.0% | 1.0 | 0.115 | 100% | 1.26 | 0.378 | 100.0% | 1.20 | 0.006 | 100% | 1.46 |
| | | 100 | 7.4 | 0.805 | 46.7% | 1.0 | 0.812 | 73.3% | 1.27 | 1.617 | 60.0% | 1.0 | 0.119 | 100% | 1.26 | 0.483 | 100.0% | 1.40 | 0.007 | 100% | 1.0 |
| LOGISTICS (673) | 10.5 | 10 | 2.9 | 3.668 | 28.1% | 1.47 | 3.658 | 54.9% | 1.66 | 7.326 | 90.2% | 2.27 | 1.201 | 99.3% | 2.98 | † | † | † | 0.563 | 55.5% | 1.24 |
| | | 30 | 8.2 | 3.416 | 27.5% | 1.07 | 3.416 | 75.0% | 1.08 | 6.832 | 90.2% | 1.2 | 1.798 | 98.6% | 1.39 | † | † | † | 0.571 | 76.4% | 1.20 |
| | | 50 | 13.4 | 3.417 | 28.1% | 1.01 | 3.409 | 91.5% | 1.05 | 6.826 | 90.8% | 1.03 | 2.545 | 98.6% | 1.29 | † | † | † | 0.599 | 86.2% | 1.10 |
| | | 70 | 18.9 | 3.799 | 28.1% | 0.96 | 3.8 | 91.5% | 0.95 | 7.599 | 92.2% | 0.99 | 3.460 | 100% | 1.13 | † | † | † | 0.608 | 96.7% | 1.05 |
| | | 100 | 26.5 | 3.786 | 31.1% | 0.97 | 3.77 | 93.4% | 0.93 | 7.556 | 93.4% | 0.93 | 4.887 | 100% | 1.0 | † | † | † | 0.615 | 100% | 1.0 |
| MICONIC (364) | 6.0 | 10 | 3.9 | 2.617 | 39.3% | 1.32 | 2.616 | 69.0% | 1.4 | 5.233 | 100.0% | 2.12 | 0.838 | 100% | 3.26 | † | † | † | 0.321 | 54.7% | 1.26 |
| | | 30 | 11.1 | 2.612 | 39.3% | 1.14 | 2.614 | 95.2% | 1.24 | 5.226 | 100.0% | 1.19 | 1.196 | 100% | 1.58 | † | † | † | 0.326 | 90.1% | 1.08 |
| | | 50 | 18.1 | 2.614 | 39.3% | 1.13 | 2.61 | 100.0% | 1.1 | 5.224 | 100.0% | 1.1 | 1.722 | 100% | 1.28 | † | † | † | 0.339 | 96.4% | 1.01 |
| | | 70 | 25.3 | 3.941 | 39.3% | 1.06 | 3.941 | 100.0% | 1.0 | 7.882 | 100.0% | 1.01 | 2.504 | 100% | 1.03 | † | † | † | 0.344 | 100% | 1.0 |
| | | 100 | 35.6 | 4.116 | 39.3% | 1.07 | 4.048 | 100.0% | 1.0 | 8.164 | 100.0% | 1.0 | 5.105 | 100% | 1.0 | † | † | † | 0.356 | 100% | 1.0 |
| ROVER (364) | 6.0 | 10 | 3.0 | 3.993 | 52.4% | 2.26 | 4.023 | 69.0% | 1.58 | 8.016 | 92.9% | 2.39 | 0.704 | 98.8% | 2.85 | † | † | † | 0.310 | 51.1% | 1.10 |
| | | 30 | 7.9 | 3.952 | 48.8% | 1.68 | 3.916 | 90.5% | 1.27 | 7.868 | 84.5% | 1.14 | 1.029 | 100% | 1.66 | † | † | † | 0.323 | 69.1% | 1.07 |
| | | 50 | 12.7 | 3.79 | 50.0% | 1.43 | 3.781 | 94.3% | 1.13 | 7.571 | 97.6% | 1.11 | 1.355 | 100% | 1.29 | † | † | † | 0.331 | 85.7% | 1.01 |
| | | 70 | 17.9 | 3.763 | 52.4% | 1.32 | 3.79 | 100.0% | 1.02 | 7.553 | 97.6% | 1.0 | 1.796 | 100% | 1.07 | † | † | † | 0.345 | 91.6% | 1.0 |
| | | 100 | 24.9 | 3.772 | 53.6% | 1.21 | 3.776 | 100.0% | 1.0 | 7.548 | 100.0% | 1.0 | 2.292 | 100% | 1.07 | † | † | † | 0.356 | 100% | 1.0 |
| SATELLITE (364) | 6.5 | 10 | 2.1 | 3.922 | 30.9% | 1.67 | 3.902 | 64.3% | 2.21 | 7.824 | 91.7% | 2.7 | 1.049 | 97.6% | 3.41 | 14.821 | 89.3% | 4.86 | 0.431 | 47.6% | 1.21 |
| | | 30 | 5.4 | 3.928 | 28.6% | 1.51 | 3.892 | 91.7% | 1.73 | 7.82 | 91.7% | 1.65 | 1.182 | 97.6% | 2.40 | 32.172 | 86.9% | 4.21 | 0.442 | 69.1% | 1.14 |
| | | 50 | 8.7 | 3.956 | 32.1% | 1.29 | 3.928 | 95.2% | 1.26 | 7.884 | 95.2% | 1.27 | 1.398 | 97.6% | 1.69 | 51.567 | 88.1% | 3.65 | 0.458 | 80.9% | 1.10 |
| | | 70 | 12.2 | 3.904 | 32.1% | 1.19 | 3.929 | 100.0% | 1.08 | 7.833 | 96.4% | 1.07 | 1.884 | 96.4% | 1.52 | 75.363 | 92.8% | 2.89 | 0.460 | 94.1% | 1.03 |
| | | 100 | 16.8 | 3.955 | 32.1% | 1.14 | 3.903 | 100.0% | 1.04 | 7.858 | 96.4% | 1.04 | 2.107 | 96.4% | 1.33 | 113.381 | 100.0% | 2.57 | 0.475 | 100% | 1.07 |
| SOKOBAN (364) | 7.3 | 10 | 3.1 | 5.883 | 22.6% | 1.19 | 5.851 | 64.3% | 1.27 | 11.734 | 67.9% | 1.27 | 3.025 | 69.1% | 4.02 | 461.701 | 67.8% | 2.98 | 0.523 | 51.1% | 1.85 |
| | | 30 | 8.7 | 5.854 | 19.1% | 1.02 | 5.73 | 89.3% | 1.02 | 11.584 | 85.7% | 1.06 | 4.429 | 89.2% | 4.10 | 370.412 | 83.3% | 3.14 | 0.531 | 55.9% | 1.21 |
| | | 50 | 14.1 | 5.911 | 21.4% | 1.04 | 5.71 | 96.4% | 1.02 | 11.621 | 90.5% | 1.0 | 7.553 | 89.2% | 4.16 | 358.028 | 82.1% | 2.27 | 0.540 | 69.1% | 1.20 |
| | | 70 | 19.8 | 5.897 | 22.6% | 1.08 | 5.653 | 100.0% | 1.01 | 11.55 | 96.4% | 1.01 | 9.112 | 89.2% | 4.17 | 353.721 | 85.7% | 1.84 | 0.554 | 86.9% | 1.08 |
| | | 100 | 35.5 | 5.849 | 25.0% | 1.07 | 5.572 | 100.0% | 1.0 | 11.421 | 100.0% | 1.0 | 12.008 | 89.2% | 4.53 | 353.183 | 85.7% | 1.03 | 0.562 | 100% | 1.0 |
| ZENO (364) | 7.5 | 10 | 2.6 | 5.474 | 34.5% | 1.33 | 5.45 | 58.3% | 1.68 | 10.924 | 82.1% | 2.62 | 1.834 | 96.4% | 3.41 | 93.917 | 66.6% | 1.63 | 0.491 | 36.9% | 1.04 |
| | | 30 | 6.7 | 5.424 | 33.3% | 1.24 | 5.449 | 86.9% | 1.35 | 10.873 | 89.3% | 1.57 | 2.528 | 88.1% | 2.11 | 88.285 | 78.6% | 2.27 | 0.504 | 60.7% | 1.02 |
| | | 50 | 10.8 | 5.005 | 33.3% | 1.2 | 5.003 | 95.2% | 1.1 | 10.008 | 91.7% | 1.1 | 3.071 | 92.8% | 1.41 | 105.814 | 91.6% | 2.56 | 0.516 | 76.1% | 1.0 |
| | | 70 | 15.2 | 4.377 | 35.7% | 1.17 | 4.321 | 100.0% | 1.0 | 8.698 | 100.0% | 1.0 | 3.986 | 96.4% | 1.13 | 125.652 | 94.1% | 2.58 | 0.522 | 90.4% | 1.0 |
| | | 100 | 21.1 | 4.378 | 35.7% | 1.18 | 4.297 | 100.0% | 1.0 | 8.675 | 100.0% | 1.0 | 4.815 | 100% | 1.07 | 168.674 | 100.0% | 1.0 | 0.530 | 100% | 1.0 |
| Average | | | | 3.927 | 30.6% | 1.202 | 3.908 | 82.7% | 1.25 | 11.761 | 92.4% | 1.440 | 2.697 | 94.7% | 2.110 | 7.834 | 63.0% | 3.464 | 0.311 | 79.7% | 1.122 |

Table 1: Goal recognition experiments at various levels of observability.

| # | $|\mathcal{G}|$ | % Obs | $|O|$ | $h$ Time | Acc % | $S$ in $\mathcal{G}$ | $h_c$ Time | Acc % | $S$ in $\mathcal{G}$ | $h_\delta$ Time | Acc % | $S$ in $\mathcal{G}$ | R&G 2009 Time | Acc % | $S$ in $\mathcal{G}$ | FG 2015 Time | Acc % | $S$ in $\mathcal{G}$ | $h_{uniq}$ Time | Acc % | $S$ in $\mathcal{G}$ |
|---|---|---|---|---|---|---|---|---|---|---|---|---|---|---|---|---|---|---|---|---|---|
| CAMPUS (516) | 2 | 25 | 3.1 | 0.627 | 53.5% | 1.0 | 0.629 | 83.0% | 1.24 | 1.256 | 87.6% | 1.12 | 0.073 | 88.3% | 1.27 | 0.713 | 79.8% | 1.33 | 0.030 | 82.1% | 1.13 |
| | | 50 | 4.5 | 0.634 | 53.5% | 1.0 | 0.634 | 92.3% | 1.19 | 1.268 | 92.3% | 1.06 | 0.076 | 89.9% | 1.26 | 0.666 | 90.6% | 1.67 | 0.031 | 78.2% | 1.02 |
| | | 75 | 6.4 | 0.625 | 53.5% | 1.19 | 0.625 | 95.3% | 1.09 | 1.25 | 94.6% | 1.09 | 0.079 | 90.6% | 1.27 | 0.655 | 94.6% | 1.79 | 0.034 | 73.6% | 1.0 |
| | | 100 | 7.5 | 0.619 | 53.5% | 1.0 | 0.62 | 95.3% | 1.22 | 1.239 | 94.6% | 1.04 | 0.084 | 89.1% | 1.22 | 0.644 | 97.7% | 1.81 | 0.037 | 72.1% | 1.0 |
| IPC-GRID (300) | 8.3 | 25 | 4.1 | 3.421 | 13.3% | 1.0 | 3.406 | 30.0% | 1.02 | 6.827 | 86.7% | 1.49 | 0.537 | 71.1% | 2.65 | 0.494 | 43.3% | 2.31 | 0.102 | 30.0% | 1.11 |
| | | 50 | 7.6 | 3.392 | 13.3% | 1.0 | 3.384 | 68.9% | 1.1 | 6.776 | 96.7% | 1.14 | 0.649 | 95.5% | 1.28 | 0.511 | 81.1% | 1.78 | 0.116 | 64.4% | 1.03 |
| | | 75 | 11.5 | 3.399 | 13.3% | 1.0 | 3.392 | 98.9% | 1.01 | 6.791 | 97.8% | 1.07 | 0.712 | 100% | 1.01 | 0.654 | 93.3% | 1.10 | 0.124 | 87.7% | 1.03 |
| | | 100 | 16.9 | 3.403 | 13.3% | 1.0 | 3.41 | 100.0% | 1.0 | 6.813 | 100.0% | 1.0 | 0.805 | 100% | 1.06 | 0.885 | 100.0% | 1.06 | 0.136 | 100% | 1.0 |
| INTRUSION (300) | 16.6 | 25 | 3.6 | 4.422 | 10.0% | 1.24 | 4.433 | 26.7% | 1.71 | 8.855 | 71.1% | 2.7 | 0.462 | 12.2% | 7.55 | Timeout | - | - | 0.208 | 53.3% | 1.72 |
| | | 50 | 6.7 | 4.457 | 10.0% | 1.12 | 4.468 | 75.6% | 1.41 | 8.925 | 96.7% | 1.33 | 0.469 | 4.4% | 8.06 | Timeout | - | - | 0.212 | 83.3% | 1.33 |
| | | 75 | 10.2 | 4.396 | 10.0% | 1.07 | 4.394 | 90.0% | 1.11 | 8.79 | 98.0% | 1.01 | 0.475 | 6.6% | 7.88 | Timeout | - | - | 0.224 | 94.4% | 1.08 |
| | | 100 | 15.1 | 4.451 | 10.0% | 1.03 | 4.44 | 100.0% | 1.0 | 8.891 | 100.0% | 1.0 | 0.476 | 10.0% | 7.76 | Timeout | - | - | 0.239 | 100% | 1.0 |
| KITCHEN (150) | 2.0 | 25 | 2.5 | 0.678 | 0.0% | 0.0 | 0.812 | 46.7% | 1.0 | 1.49 | 73.3% | 1.69 | 0.139 | 71.1% | 1.57 | 0.381 | 53.3% | 1.33 | 0.081 | 88.8% | 2.55 |
| | | 50 | 4.8 | 0.681 | 0.0% | 0.0 | 0.809 | 51.1% | 1.04 | 1.49 | 55.6% | 1.33 | 0.135 | 57.7% | 1.42 | 0.410 | 51.1% | 1.22 | 0.084 | 64.4% | 1.71 |
| | | 75 | 7.3 | 0.682 | 0.0% | 0.0 | 0.819 | 48.9% | 1.02 | 1.501 | 53.3% | 1.33 | 0.138 | 57.7% | 1.31 | 0.426 | 53.3% | 1.20 | 0.090 | 57.7% | 1.66 |
| | | 100 | 11.0 | 0.683 | 0.0% | 0.0 | 0.811 | 73.3% | 1.27 | 1.494 | 53.3% | 1.13 | 0.144 | 60.0% | 1.46 | 0.538 | 73.3% | 1.26 | 0.093 | 66.6% | 1.13 |
| Average | | | | 2.286 | 19.2% | 0.779 | 2.318 | 73.5% | 1.158 | 4.604 | 84.6% | 1.283 | 0.341 | 62.8% | 2.998 | 30.436 | 76.0% | 1.488 | 0.115 | 74.8% | 1.281 |

Table 2: Goal recognition experiments with noisy observations at various levels of observability.

stantially superior to the state-of-the-art over a large dataset. Importantly, the values of the operator-counting constraints we compute for each of the heuristics can be used as explanations for recognized goals.

The techniques described in this paper use a set of simple additional constraints in the ILP formulation to achieve substantial performance, so we expect substantial future work towards further goal recognition approaches and heuristics that explore more refined constraints to improve accuracy and reduce spread, as well as deriving a probabilistic approach using operator-counting information. Examples of such work include using the constraints to force the LP to generate the counterfactual operator-counts (i.e., non-compliant with the observations) used by the R&G approach, or, given an estimate of the noise, relax the observation constraints to allow a number of observations to not be included in the resulting operator-counts.

**Acknowledgements:** This study was financed in part by the Coordenação de Aperfeiçoamento de Pessoal de Nivel Superior – Brasil (CAPES) - Finance Code 001. Felipe acknowledges support from CNPq under project numbers 407058/2018-4 and 305969/2016-1, as well as FAPERGS process number 18/2551-0000500-2.

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
