# OpenReview forum: "Robust Goal Recognition with Operator-Counting Heuristics"
_icaps-conference.org/ICAPS/2019/Workshop/XAIP — XAIP 2019_

### Official Review · AnonReviewer1 · 2019-04-24
**Nice paper on improved goal recognition but why XAIP?**

**Rating:** 2
**Confidence:** 2

**Review:**

This is a nicely done work on improving goal recognition in the planning-based framework, cleverly leveraging operator-counting heuristics.

I don't have anything to criticize on that content -- other than, why was it submitted to XAIP?

My best hypothesis is, the authors wanted to submit to some ICAPS workshop, browsed the list, and deemed XAIP to be closest.

Which in itself could still be fine. But the topic match is far from obvious, and the authors do not even make an attempt to discuss it. It cannot be the task of the reader to figure out how this is relevant to XAIP. (Actually it seems the authors submitted as-is, not even taking proper care in reformatting to AAAI style; in some places Section refs are broken because there is no section numering anymore; but this is just an aside)

The best I can come up with personally is that goal recognition can be construed as related to XAI as a prerequisite for understanding another agent's behaviour. But then, this is like a non-cooperative setting of XAI, where the agent one is trying to understand is not actually trying to be understood. Maybe this is relevant in some application, I don't know, but XAI as the term is usually understood means the opposite, trying to make an AI agent understandable to its environment/its users.

In any case, making sense of this can't be the task of the reader. And with so many submissions to XAIP we can't just accept anything, we need tp focus. So I lean to reject. Perhaps the authors could be invited to a panel on viewpoints on XAIP, if such a panel is oplanned (and if they are indeed willing to argue a connection to goal recognition).

---

### Official Review · AnonReviewer2 · 2019-05-08
**Relevance to XAIP not clear**

**Rating:** 2
**Confidence:** 1

**Review:**

I am not sure if the paper is relevant to the XAIP Workshop. I suppose goal recognition can be seen as a way of trying to "explain" what the goal of an agent is from observations. But the authors made no attempt to make that stance. Even if that is the case, there is plenty of existing work [1] where the agent tries to make the goal or plan recognition task easier for the observer.

If there is a case to be made for operator count heuristics specifically for this purpose, I would love to hear it.

[1] Explicability? Legibility? Predictability? Transparency? Privacy? Security? The Emerging Landscape of Interpretable Robot Behavior. Tathagata Chakraborti, Anagha Kulkarni, Sarath Sreedharan, David Smith, Subbarao Kambhampati.

[2] Also this might be interesting to the authors (at the intersection of XAIP and landmarks): https://arxiv.org/abs/1903.08218

---

> ### Author Response · Authors · 2019-05-14
> **Relation to the workshop and suggested references**
>
> While I appreciate the time the reviewer took to read the paper, I find this review strongly objectionable for two key reasons:
> 1. Goal reasoning is one of the keywords of the workshop, and not much explanation as to what this means exactly is given in the workshop description. We submit that goal recognition is a form of goal reasoning, so being able to quickly and accurately identify the goal of an agent given a partial plan is clearly useful when trying to explain behaviour online. While the paper itself might not have this laid out explicitly, this is a minor presentation issue which we can easily rectify.
> 2. The suggested references are both only available in ArXiV rather than major conferences, and, while I regularly try to keep up with preprint publication, the volume (of even filtered articles) is too large to meaningfully monitor. Moreover, while it would be feasible (but hard) to find and use the material in reference 1, reference 2 was posted to ArXiV less than a month before the deadline of this workshop.
>
> In summary, this review dismisses our paper outright without actually referring to any of the paper content while making unreasonable demands about the kind of background we should include in the paper.

---

> > ### Comment · AnonReviewer2 · 2019-05-14
> > **clarifications**
> >
> > The references are speculative from me in trying to make sense of where goal recognition and landmarks can possibly fit into the XAIP theme.  My objection was that the paper makes no attempt to make that connection and so I am left to make them myself.
> >
> > Goal reasoning usually refers to the argumentation over which goal to achieve and not goal recognition. Goal recognition exists as a field, but that connection to XAIP needs to be made for a XAIP paper. I can only comment with respect to what was there in the paper, and I could not find any connection to the cause of explaining behavior. Your clarification is appreciated.

---

> > > ### Author Response · Authors · 2019-05-14
> > > **Reservations**
> > >
> > > So, given the clarifications in the review process, do you agree that the rating and confidence of your review might need a revision? As I argued, while the connection to the area might not be explicit in the paper, is rather straightforward.

---

> > > > ### Comment · AnonReviewer2 · 2019-05-14
> > > > **clarifications**
> > > >
> > > > Sure, though both reviews will be taken into consideration by the chairs before a decision is made.
> > > >
> > > > Just to be clear, this is not a venue for argument. The references were meant to help you make that connection to explainable planning. One of them is a survey that looks at a whole spectrum of works on how the cause of goal recognition can be helped. The other one I thought you might find interesting.

---

### Decision · Program_Chairs · 2019-05-15

**Decision:**

Accept

**Comment:**

While the reviewers view this paper critically, the authors are correct in that goal reasoning is part of the CFP for this workshop. Furthermore, in the spirit of making the workshop a venue for discussion and feedback, we decided to reject only those papers with strong reject votes.

Please address the review criticism as best possible for the final paper version and its presentation at the workshop. In particular, please carefully discuss the relation to/links to XAIP, and the XAIP literature. Looking forward to discuss your work at the workshop!